# The Contribution of Microglia and Brain-Infiltrating Macrophages to the Pathogenesis of Neuroinflammatory and Neurodegenerative Diseases during TMEV Infection of the Central Nervous System

**DOI:** 10.3390/v16010119

**Published:** 2024-01-13

**Authors:** Ana Beatriz DePaula-Silva

**Affiliations:** Department of Pharmacology and Toxicology, College of Pharmacy, University of Utah, Salt Lake City, UT 84112, USA; bea.silva@pharm.utah.edu

**Keywords:** TMEV, picornavirus, neurotropic, neuroinflammation, multiple sclerosis, epilepsy, macrophages, microglia, cytokines, CNS

## Abstract

The infection of the central nervous system (CNS) with neurotropic viruses induces neuroinflammation and is associated with the development of neuroinflammatory and neurodegenerative diseases, including multiple sclerosis and epilepsy. The activation of the innate and adaptive immune response, including microglial, macrophages, and T and B cells, while required for efficient viral control within the CNS, is also associated with neuropathology. Under healthy conditions, resident microglia play a pivotal role in maintaining CNS homeostasis. However, during pathological events, such as CNS viral infection, microglia become reactive, and immune cells from the periphery infiltrate into the brain, disrupting CNS homeostasis and contributing to disease development. Theiler’s murine encephalomyelitis virus (TMEV), a neurotropic picornavirus, is used in two distinct mouse models: TMEV-induced demyelination disease (TMEV-IDD) and TMEV-induced seizures, representing mouse models of multiple sclerosis and epilepsy, respectively. These murine models have contributed substantially to our understanding of the pathophysiology of MS and seizures/epilepsy following viral infection, serving as critical tools for identifying pharmacological targetable pathways to modulate disease development. This review aims to discuss the host–pathogen interaction during a neurotropic picornavirus infection and to shed light on our current understanding of the multifaceted roles played by microglia and macrophages in the context of these two complexes viral-induced disease.

## 1. Introduction

More than 100 viruses are capable of infecting the central nervous system (CNS), leading to altered CNS function. An increased risk for many neurological disorders is often observed following viral encephalitis. The *Picornaviridae* family, commonly referred to as picornaviruses, are small, non-enveloped positive-sense single-stranded RNA viruses [1]. This family comprises a growing list of 40 genera that includes *Enterovirus*, *Rhinovirus*, *Hepatovirus*, *Cardiovirus*, *Aphthovirus*, and many others [2]. Picornaviruses are amongst the most prevalent human pathogens and can cause a wide range of diseases in both animals and humans, spanning from asymptomatic conditions to severe life-threatening diseases [3]. Certain picornaviruses, such as poliovirus and coxsackievirus, both belonging to the *Enterovirus* genus, exhibit a neurotropic nature [3,4,5]. They can enter and establish infection in the CNS of humans, potentially resulting in meningitis or encephalitis. The CNS has a unique and complex structure with a restricted regeneration capacity. The perturbation of the CNS homeostasis, induction of neuroinflammation, neuronal damage, and neuron degeneration often occur as a result of a CNS viral infection and are closely associated with the development of neuroinflammatory and neurodegenerative diseases such as multiple sclerosis (MS) [6,7,8,9,10,11], Alzheimer’s [10,12,13], epilepsy [14,15,16,17,18,19,20,21,22,23,24,25], and others [26,27,28,29,30]. In fact, viral infections are known or suspected to play a role in the onset and progression of various neurological disorders. Despite recent advancements, the mechanisms underlying how a viral infection contributes to neuroinflammation and neurodegeneration are complex and not entirely understood.

The use of preclinical animal models is a crucial instrument to study the mechanisms of disease initiation, development, and progression. This approach can reveal novel therapeutic targets for preventing, interrupting, and reversing disease outcomes, ultimately improving the quality of life of patients suffering from these diseases. This review will focus on using the picornavirus Theiler’s murine encephalomyelitis virus (TMEV) in two preclinical models related to neurological disorders: MS and epilepsy (Figure 1). The primary aim of this work is to describe the immune response to TMEV during CNS viral infection (Figure 2), explain each model, and, lastly, highlight the different roles played by microglia and macrophages in each mouse model, focusing on its contribution to disease development.

## 2. An Overview of TMEV

Based on the nucleotide sequence and organization of the genome, TMEV belongs to the *Cardiovirus* genus within the *Picornaviridae* family. The genus *Cardiovirus* includes TMEV, encephalomyocarditis virus (EMCV), and Saffold virus (SAFV). It is noteworthy that SAFV is the only cardiovirus known to infect humans [31,32]. TMEV is an enteric pathogen of mice, and in natural infections, TMEV can be found in the intestinal mucosa and fecal matter (reviewed in [33]). Viral transmission occurs via the oral–fecal route [34], and infected mice can shed TMEV through their intestinal contents. While the infection of TMEV among wild mice occurs, these mice rarely become symptomatic. Nevertheless, while the spread of TMEV from the intestinal tract to the CNS is infrequent, when it occurs it can result in severe neurological disorders. TMEV was first identified by Max Theiler in 1937 when he isolated the TMEV original strain (TO) from the CNS of mice exhibiting flaccid hind leg paralysis [35,36,37]. Subsequently, another strain, known as Daniels (DA), was isolated by Joan Daniels in 1952 from spontaneously paralyzed mice and was shown to cause chronic inflammatory demyelinating disease in the spinal cord of mice [38]. Experimentally, TMEV infection of the CNS results in distinct neurological manifestations depending on the mouse’s genetic background and the virus strain (reviewed in [16,39,40]).

Based on neurovirulence, TMEV can be categorized into two subgroups: highly neurovirulent (George Davis 7—GDVII) and low neurovirulent (Theiler’s original—TO). Within the GDVII subgroup are the GDVII and FA strains. The DA, BeAn 8386 (BeAn), TO4, Yale, WW, and 4727 strains are included in the TO subgroup. The highly neurovirulent strains induce acute fatal encephalomyelitis in mice within one-to-two weeks post-infection (pi) [41]. In contrast, the low neurovirulent strains cause different diseases depending on the mouse strain and route of inoculation (Figure 1). The intracerebral (IC) infection of Swiss Jim Lambert (SJL) mice with low neurovirulent strains, such as DA or BeAn, results in a biphasic disease characterized by an acute polioencephalomyelitis and a later chronic progressive inflammatory demyelinating disease, which is associated with viral persistence in the CNS (Figure 1A) [41,42,43]. Notably, the DA and BeAn strains show differences in neurovirulence. A comparison between DA and BeAn sequences revealed genetic differences in the capsid proteins (VP1 and VP3), in the nonstructural protein 3A, and in the polymerase (3D) [44,45]. However, the specific genetic variations responsible for the difference in neurovirulence remain to be determined. In contrast, an IC infection of C57/BL6J (BL6) mice with the DA strain results in acute encephalitis accompanied by behavioral seizures and a later development of recurrent spontaneous seizures, indicative of epilepsy (Figure 1B) [33,46]. Thus, TMEV infection of SJL mice serves as a mouse model of TMEV-induced demyelinating disease (TMEV-IDD), while TMEV infection of BL6 mice is used as a mouse model of viral-induced epilepsy [16,20,39,40,47]. These models are valuable tools for studying viral-induced progressive MS and viral-induced temporal lobe epilepsy (TLE), respectively (Figure 1). The difference in disease development in BL6 compared to SJL mice is, in part, due to the type and intensity of the immune response elicited by these two distinct strains of mice. Some of these differences are highlighted later in this review. It is important to note that TMEV infection can also result in myocarditis when injected intraperitoneally in C3H mice [48,49,50]. This disease model is beyond the scope of this review and is not discussed herein. The following sections will focus primarily on the DA-TMEV strain, referred to as TMEV henceforth.

## 3. TMEV Infection

The exact mechanism of TMEV cell entry is not fully understood. It is suggested that TMEV enters the cells by binding to a specific receptor expressed at the cell surface. While TMEV is known to target sialic acids in the gastrointestinal tract in in vitro settings [50,51,52], the receptors used by TMEV during an in vivo CNS viral infection are yet to be determined. Viral binding to the host cell surface receptor leads to a viral uncoating [1,31,53,54], allowing the release of the viral genome into the cytoplasm. Based on studies of other members of the picornavirus family, it is suggested that TMEV may utilize receptor-mediated endocytosis to enter the host cell. This pathway could explain how TMEV infection activates endosomal Toll-like receptors (TLRs), such as TLR3 and TLR7 [54,55,56]. The TMEV genome is released from the endosomal compartment into the cytoplasm, where the viral RNA serves as a template for viral replication and translation.

The TMEV genome is 8.1 kb in size, flanked by 5′ and 3′ untranslated regions (UTRs) and a polyA tail at the 3′ (Figure 3) [54]. The positive-sense RNA is transcribed into its complementary negative strand, resulting in the formation of a double-stranded RNA (dsRNA), which is used as a template to generate progeny viral RNA. Furthermore, the positive-sense RNA acts as a template for protein translation. An internal ribosome entry site (IRES) in the 5′ UTR is used to translate the RNA genome into a polyprotein. This polyprotein is subsequently proteolytically processed by the 3C protease to generate 12 proteins, including structural (VP1, VP2, VP3, and VP4) and nonstructural (2A, 2B, 2C, 3A, 3B, 3C, 3D, and L) proteins (Figure 3) [57,58]. An alternative open reading frame (ORF) is used to express an additional 17 kDa protein called L*, which is present in the TO subgroup but not in the GDVII subgroup [59,60]. Structural proteins are essential for viral assembly, while nonstructural proteins play a role in many processes required for effective viral infection, such as the hijacking of the host machinery, the processing of the viral polyprotein and translation, viral genome replication, and counteracting the host immune response [58]. Newly synthesized virions are released from the infected cell via cell lysis, indicative of a lytic infection.

## 4. Innate Immune Response by Microglia and Macrophages during CNS Viral Infection

A characteristic of neuroinflammation induced by a CNS infection involves the presence of reactive neuroglia such as microglia, oligodendrocytes, NG2-glia, and astrocytes, and the recruitment of peripheral leukocytes, including monocytes/macrophages, neutrophils, and lymphocytes, which are critical to control viral infections [2,16,39,40,42,61,62,63,64,65,66,67,68]. Microglia are myeloid cells commonly referred to as macrophages of the brain. Although microglia and monocytes/macrophages share many characteristics, they each have a specific ontogeny. Microglia originate from the embryonic yolk sac and infiltrate the CNS during embryogenesis. These cells have self-renewal properties and are independent of the adult hematopoiesis [69,70,71]. In contrast, peripheral macrophages/monocytes are derived from adult bone marrow hematopoietic stem cells, have a shorter lifespan, and are continuously replenished throughout postnatal life [71,72,73,74]. Peripheral blood monocytes are found in the periphery and do not normally migrate into the CNS through a healthy and intact blood–brain barrier (BBB). However, under pathological conditions, monocytes can infiltrate the CNS and differentiate into macrophages [75]. Microglia and macrophages are highly engaged in the neuroinflammatory process during viral encephalitis [76,77] and are important players in the induction of the innate and adaptive immune response.

Under homeostasis, microglia remain quiescent, highly ramified, and continually engage in immune surveillance that is crucial for the development and maturation of the CNS microenvironment. Upon changes in CNS homeostasis, reactive microglia undergo both functional and morphological changes, adopting an amoeboid shape, less ramified, and increased cell body size [62,70,78]. Like macrophages, microglia exhibit a high degree of plasticity, and, depending on the environmental stimuli and spatial localization, they can adopt distinct reactive states with a large and complex spectrum of phenotypes [70,72,78,79]. Despite the substantial recent advances in single-cell sequencing and multi-omic techniques, which have provided a better understanding of microglial heterogeneity and functional states, defining specific features and nomenclature for these various states remains a challenge. Nonetheless, researchers are actively working towards addressing this issue [80].

Both microglia and macrophages are dynamic cells and can adopt a reactive inflammatory or reactive anti-inflammatory state, depending on the stimuli. Inflammatory microglia promote inflammation by secreting pro-inflammatory cytokines, while anti-inflammatory microglia secrete anti-inflammatory mediators to reduce inflammation and promote tissue healing and repair. During a CNS viral infection, microglia can become reactive by recognizing pathogen-associated molecular patterns (PAMPs) on viruses, as well as damage-associated molecular patterns (DAMPs), such as extracellular adenosine 5′-triphosphate (ATP) released by infected or damaged neurons, through P2 purinergic receptors [78,81,82,83] (Figure 2). This recognition triggers the activation of microglia towards an inflammatory reactive state and increases microglial phagocytic activity, leading to the migration of microglia to the site of injury and the generation of antiviral response, including the induction of type-1 interferon (IFN-I) and the secretion of proinflammatory cytokines (Figure 2). Inflammatory microglia also facilitate the recruitment of peripheral leukocytes, including macrophages, by secreting chemokines such as chemokine (C-C motif) ligand 2 (CCL2), which is recognized by the C-C chemokine receptor type-2 (CCR2) expressed on macrophages, allowing for a macrophage’s migration towards a higher local concentration of CCL2 [84,85]. Once inside the CNS, reactive inflammatory macrophages and microglia play pivotal roles in mounting an inflammatory antiviral immune response (Figure 2). As a consequence, this response can induce neuroinflammation, activate NG-2 glia and astrocytes, and amplify the breakdown of the BBB, contributing to the development of CNS diseases (Figure 2).

Differentiating between microglia and macrophages in the setting of CNS inflammation, although crucial, has been challenging. These cells share similarities in both their morphology and reactive state. Flow cytometry has been a valuable tool for distinguishing microglia from macrophages, primarily relying on the CD45 level of expression by these cells. CD45 is highly expressed in all nucleated cells from the hematopoietic origin, but microglia have been shown to express low-to-intermediate levels of this protein at the cell surface. Therefore, microglia are characterized as CD45^low/int^Cd11b^+^, while macrophages are identified as CD45^hi^Cd11b^+^ [16,18,39,40,76]. While this identification method is well accepted and used, it remains a subject of controversy. Several studies have raised doubts about the reliability of CD45-based differentiation, as some studies suggest that microglia can upregulate CD45 expression during neuroinflammation, becoming indistinguishable from macrophages [86]. A recent study proposes using Ly6C/G expression as an alternative means to discriminate between microglia and macrophages during neuroinflammation [65].

While differences in the level of CD45 expression are used to study these two populations by flow cytometry, when it comes to direct visualization, such as through immunohistochemistry, a specific marker is necessary. Numerous studies have aimed to identify new proteins unique to either microglia or macrophages. Notably, markers such as the purinergic receptor P2RY12 and the transmembrane protein 119 (TMEM119), initially described as being specific to microglia, failed to be useful during neuroinflammation [87,88]. This is due to their downregulation upon microglial activation, rendering them ineffective as a marker to differentiate microglia from blood macrophages that infiltrate the brain [79,89,90,91,92]. Despite advancements in single-cell sequencing techniques, the challenge of the precise discrimination between microglia and macrophages remains, and genetic manipulation may offer a more reliable approach. Although CX3CR1 is expressed in both microglia and macrophages, the high turnover rate of macrophages (approximately 3 weeks) allows for the use of CX3CR1-Cre lines to selectively label microglia [76]. However, it is important to note that studies have indicated CX3CR1 expression in both neurons and astrocytes [93,94]. In the context of neuroinflammation, CCR2-Red Fluorescence Protein (RFP) mice have shown the specific labeling of macrophages that infiltrate the brain. Nevertheless, there is an ongoing discussion about CCR2 being expressed by cells other than macrophages [95]. Consequently, using caution in studying microglia- and macrophage-specific roles is crucial.

### 4.1. The Role of INF-I Response to TMEV Infection

The recognition of viral antigens and the initiation of the innate immune response are critical steps in the host’s defense against viral infections. PAMPs are conserved structures among microbial species, such as dsRNA, ssRNA, and lipopolysaccharides (LPS), and are recognized by the pattern recognition receptors (PRRs) encoded by the host’s germline. PRRs are expressed in a variety of cells and are classified into five distinct groups: TLRs, nucleotide oligomerization domain (NOD)-like receptors (NLRs), retinoid acid-inducible gene-I (RIG-I)-like receptors (RLRs), C-type lectin receptors (CLRs), and absent in melanoma-2 (AIM2)-like receptors (ALRs) [55,56,96,97,98,99,100,101]. Upon the recognition of the PAMPs by the PRRs, downstream signaling pathways are activated, playing a vital role in antiviral responses. This activation results in various effects, including the secretion of cytokines, chemokines, growth factors, and other molecules that impact many processes involved in innate and adaptive immune responses. One significant outcome is the expression of IFN-I [56,101]. Interestingly, the expression of PRRs is differentially distributed in the CNS, which may contribute to diverse responses and specific local outcomes.

The induction of IFN-I is a hallmark feature of the immune response to viral infections, with IFNα and IFNβ being the primary cytokines in this family [97,101]. These cytokines bind to the type-I interferon receptor, comprised of two transmembrane subunits: IFNAR1 and IFNAR2. The binding of IFN-I to its receptor triggers the activation of Janus-activated kinases (JAKs) and the phosphorylation of signal transduction and transcription activation (STAT)1 and STAT2 proteins. This leads to the formation of the STAT1-STAT2-IRF9 (IFN-regulatory factor 9) complex, which translocates to the nucleus and binds to the IFN-stimulated response elements (ISREs), resulting in the expression of numerous IFN-stimulated genes (ISGs) with pro- and anti-inflammatory roles [97,101,102].

Secreted IFN-I, such as IFNα and IFNβ, can act in an autocrine manner by binding to the cell surface of IFNAR1 and IFNAR2 on the same cell that produced and secreted them or in a paracrine manner by acting on neighboring cells [103]. Both pathways induce an antiviral state to inhibit viral replication and modulate both innate and adaptive immune responses [103]. IFN-I is known to impact the natural killer (NK) cell-mediated cytotoxicity [104], increase the secretion of IFN-γ by T cells [105], promote CD4^+^ T cell T helper (Th)1 immune responses through the activation of dendritic cells, increase the expression of major histocompatibility complex (MHC) class I and costimulatory molecules presented on antigen-presenting cells (APCs), promote CD8^+^ T cell proliferation and survival, and modulate B cell response [97,106]. Furthermore, studies have also shown that the induction of IFN-I is implicated in the modulation of the BBB permeability and tight junction formation [107] through several mechanisms, including the secretion of the anti-inflammatory cytokine interleukin (IL)-10 [108].

During TMEV infection, the virus is found in the endosome before being released into the cytoplasm, where viral replication and translation occur. Within the endosomal compartment, TLR3, 7, and 9 are present and recognize dsRNA, ssRNA, and dsDNA, respectively [56]. Consequently, during TMEV infection, TLR7 senses endosomal ssRNA, signaling to MyD88 and activating the translocation of IRF7 into the nucleus, leading to the expression of IFNα/β genes. Moreover, reports suggest that TMEV dsRNA replication intermediates can also be found in the endosome, which can be sensed by TLR3. The engagement of dsRNA and TLR3 activates the Toll/IL-1R domain-containing adaptor-inducing IFN-β (TRIF)-dependent pathway, culminating in IRF3 and NF-κB (nuclear factor kappa light-chain enhancer of activated B cells) activation. Once in the cytoplasm, both TMEV ssRNA and dsRNA can be recognized by the melanoma differentiation-associated gene 5 (MDA5) RLR, which then recruits mitochondrial antiviral signaling (MAVs). This activates IRF3/7 and NF-κB, triggering interferon and inflammatory responses [98]. Viral dsRNA in the cytoplasm can also be detected via the IFN-induced dsRNA-dependent protein kinase (PKR), a eukaryotic initiation factor 2α (eIF2α) kinase involved in the translation machinery. Following binding to viral dsRNA, PKR undergoes autophosphorylation, leading to its activation and the subsequent phosphorylation of eIF2α. This phosphorylation event inhibits the translation of both viral and cellular mRNA [109,110]. PKR activation also leads to the NF-κB signaling pathway and the expression of inflammatory molecules. Notably, in SJL mice, the excessive activation of the NF-κB pathway has been suggested to promote the expression of anti-apoptotic proteins, such as B-cell lymphoma (Bcl)-2 and Bcl-xL, preventing the apoptosis of infected cells and promoting viral replication and persistence [111].

The production of oligoadenylate synthase (OAS) is induced upon the activation of the INF-I response. OAS becomes activated upon binding to cytosolic dsRNA, a product of TMEV replication. Activated OAS converts ATP into 2′-5′oligoadenylates (2-5A), which inhibits protein synthesis by activating RNase L, which cleaves both viral and cellular ssRNA, decreasing viral replication and triggering cell apoptosis [112,113,114]. Also, the generation of short RNA fragments amplifies the IFN-immune response [115].

### 4.2. TMEV Counteraction of the Interferon Response

Viruses have evolved diverse strategies to avoid host immune recognition, and for TMEV to efficiently replicate and persist within infected cells, counteracting numerous cellular host factors is critical. Two key TMEV accessory proteins, L and L* (Figure 3), play pivotal roles in interfering with the host’s innate immune response, thereby contributing to viral persistence [54]. TMEV L protein prevents the IFN response by blocking IRF3 translocation into the nucleus and inhibiting IRF3 dimerization [116,117,118]. Moreover, the L protein also prevents the export of mRNA from the nucleus, impairing the production of IFN-I transcripts [116]. During a viral infection, the formation of stress granules leads to eIF2α phosphorylation by PKR, thereby inhibiting the host’s translation machinery. Notably, the TMEV L protein prevents stress granule formation in infected cells [119]. The L* protein from TMEV is a 17 kDa protein expressed from an alternative ORF that facilitates the in vitro infection of macrophages and in vivo persistence (Figure 3). The TMEV L* protein inhibits 2-5A from binding to RNase L, antagonizing RNase L antiviral activity [112,120]. Furthermore, the TMEV 3C protease plays a critical role in modulating the host immune response. It cleaves the immune RNA sensor MDA5, preventing an IFN response during the TMEV infection [121]. In other picornaviruses, 3C has been shown to modulate the IFN response by targeting the RIG-I and NF-κB pathways, thereby affecting the induction of the IFN response [54].

## 5. Multiple Sclerosis

Worldwide, 2.8 million people live with MS [122]. While MS exhibits distinct geographic prevalence, with higher incidences in Europe and North America and lower rates in sub-Saharan Africa and East Asia, its overall prevalence is increasing globally [122]. In only 14% of the reporting countries, MS incidence has remained stable or decreased. A substantial growth in the pediatric onset of MS has also been reported [122,123]. While the reason for this increase is currently unknown, genetic and environmental factors may be associated with this trend [124]. MS is a cell-mediated chronic and progressive neuroinflammatory and neurodegenerative autoimmune disease of the CNS characterized by inflammatory demyelination, axonal damage, and progressive neurological dysfunction [125,126,127]. The loss of the myelin sheath (demyelination), that insulates the axons of neurons, and the death of myelin-producing oligodendrocytes (OPCs) affect neuronal signal conduction during action potential propagation. MS symptoms include motor, sensory, and cognitive dysfunction [128], such as tremors, weakness, loss of vision, vertigo, dementia, and paralysis. MS prevalence is two-to-three times higher in women than men and is the primary cause of nontraumatic disabilities in young adults [129]. While disease-modifying therapies are available, no curative treatment has been identified. Although the pathogenesis of MS is complex and the etiology of this disease remains largely unknown, genetic and epidemiological studies suggest that genetics, lifestyle, and environmental factors, such as smoking, obesity, and viral infections, influence MS initiation, exacerbation, and progression in susceptible individuals [130]. Genes associated with the immune response, especially in the human leukocyte antigen loci, are significantly involved in the development of MS. However, genetic predisposition alone seems not to be responsible for disease development [131], as demonstrated by monozygotic twin studies [132].

For many decades, viruses have been suspected as a trigger of MS. Many viruses can infect the CNS, resulting in acute or delayed neuropathology, and epidemiological studies support a link between viral infections and MS [10,133,134]. Epstein–Barr virus (EBV), human herpesvirus 6 (HHV-6), varicella–zoster virus, cytomegalovirus, John Cunningham virus, and human endogenous retroviruses have been implicated in MS development [6,135,136,137,138,139], and rhinovirus, enterovirus, and influenza have been linked to MS exacerbation and relapses in patients [140,141]. However, the mechanisms of how the immune response induced by a viral infection results in demyelination remain a significant gap in our knowledge.

Many mechanisms have been proposed to explain the correlation between viral infection and the development of autoimmune diseases. Recent studies demonstrated that EBV-infected individuals have a 30-fold increase in the chance of developing MS compared to uninfected individuals [7]. EBV infection is highly prevalent worldwide, infecting more than 90% of the population. Certain EBV antibodies are cross-reactive to self-proteins and highly associated with MS [6,135]. Although this work identified EBV as a prerequisite for MS development, it does not explain why some individuals with the cross-reactive antibody do not show signs of disease development. Also, despite high EBV prevalence, only a few individuals develop MS, suggesting other mechanisms play a role in disease development. Importantly, lifelong viral persistence is a common feature of all the viruses proposed to be a potential trigger of MS [139], and therapies with INFβ, a disease-modifying therapy for MS, also support the role of viral infections in MS. Thus, the use of animal models is critical to study the mechanisms of the viral infection triggering neuroinflammation and demyelination.

### 5.1. TMEV-IDD, a Model of Viral-Induced Progressive MS

The innate immune response plays a pivotal role in responding to TMEV infection and in the initiation and progression of MS. IC infection with TMEV results in TMEV-IDD in SJL mice, but not in BL6 mice (Figure 1). This is attributed to viral persistence in SJL, whereas BL6 mice can effectively clear the viral infection [20,35,36,47,51]. The capacity of TMEV to persist in microglia, astrocytes, and brain-infiltrating macrophages [35,51,142,143] and their role in initiating and maintaining an antiviral immune response place these cells as important players in TMEV-IDD (Figure 1A).

TMEV-IDD is a well-established and the most used mouse model of viral-induced demyelinating disease [20,36,39,40,42,43,47,144,145,146]. SJL mice IC-infected with TMEV develop a persistent viral infection within the white matter that lasts throughout the lifespan of the mouse, leading to spinal cord demyelination (Figure 1A) [36]. This is a biphasic disease characterized by a mild acute grey matter polioencephalomyelitis phase between 3 and 14 days pi and a late, chronic progressive white matter demyelinating phase that appears 40–60 days pi, and is characterized by hindlimb flaccid paralysis, which is associated with spinal cord inflammation, demyelination, and axonal damage, with the disease progressing throughout the animal’s life span (reviewed in [20,39,40,47]). TMEV-IDD is associated with chronic neuroinflammation and autoimmune T-cell response development against myelin peptides [147,148].

The presence of active demyelinating lesions in the brain and spinal cord is a pathologic hallmark of MS in human patients [149] and presents as CNS-compartmentalized inflammation comprised of infiltrating immune cells such as macrophages, dendritic cells, T and B cells, and reactive glia [150,151,152,153,154]. These features are also observed during spinal cord demyelination in TMEV-IDD [40,144]. Thus, this mouse model recapitulates many pathological changes observed in patients with primary progressive MS, and it provides a unique opportunity to study the pathological mechanisms driving demyelination in response to viral infection.

During the acute phase of the infection, TMEV is found predominantly in neurons in the grey matter, such as the CA1 and CA2 regions of the hippocampus and cerebral cortex, and CNS inflammation (encephalitis) is characterized by the infiltration of immune cells such as macrophages, Th1 CD4^+^ T cells, CD8^+^ cytotoxic T lymphocytes (CTLs), and B cells; plasma cells are also observed (reviewed in [20,39,40,146]). In contrast to BL6 mice, in SJL-infected mice, most of the hippocampal neurons are preserved, showing mild hippocampal sclerosis [155,156]. This variation in hippocampal sclerosis is not attributed to differences in viral titer or viral tropism between BL6 and SJL mice. Instead, it is potentially a consequence of the varying intensity of the innate immune response elicited by these two distinct mouse strains [156,157,158]. Interestingly, during the acute phase of TMEV infection in SJL, a transient increase in IL-10-related genes is observed [159]. IL-10 is an anti-inflammatory cytokine that plays a critical role in controlling the immune response and inflammation. The inhibition of the IL-10 signaling in TMEV-infected SJL resulted in enhanced neuronal loss and hippocampal damage, emphasizing, once again, the significance of the amplitude of the immune response in neuronal damage [159]. 

A TMEV-specific adaptive immune response is induced to clear the virus, but a low level of TMEV persists. Through axonal transport or hematogenous routes, TMEV spreads from the grey matter into the white matter of the brain and spinal cord [160,161]. During the chronic phase, TMEV can be found in oligodendrocytes, astrocytes, microglia, and infiltrating macrophages in the white matter of the spinal cord [35,142,162,163], where it persists throughout the life of the animal. Microglia, macrophages, and astrocytes are believed to be the primary reservoir for TMEV (Figure 1A), although it remains controversial which of these cells are permissive for TMEV viral replication in infected mice [142,164,165].

Mechanisms such as bystander activation, molecular mimicry, and epitope spreading have been proposed to explain how viral infections lead to demyelinating disease [166,167,168], and chronic CNS inflammation due to the persistent viral infection is believed to be a significant driver of the immune-mediated demyelination [42,144,145,169,170]. During TMEV infection of the CNS, the continuous activation of innate and adaptive immune responses and peripheral immune cell infiltration into the CNS influences the pathogenesis of the demyelinating disease [64,171]. Demyelinating lesions in the spinal cord are in areas of active inflammation characterized by the presence of activated T cells, plasma cells, microglia, and macrophages [47,172]. MS is a T-cell-mediated disease, and innate myeloid cells play a critical role in orchestrating the adaptive immune response and, thus, are central players in the development of autoimmunity.

### 5.2. The Role of Microglia and Infiltrating Macrophages in Demyelination during TMEV Neurotropic Infection

CNS myeloid cells, such as the resident phagocytic microglia cells and border-associated macrophages (BAMs), are generally present in the healthy CNS. In contrast, blood-derived myeloid cells, such as macrophages, are found in the periphery and are physically isolated from the CNS by the BBB under normal physiological conditions. However, peripheral macrophages can infiltrate the brain parenchyma in various pathological situations, including viral encephalitis (Figure 2). As previously mentioned, both microglia and macrophages can assume either an inflammatory or anti-inflammatory dynamic reactive state. Inflammatory reactive microglia promote inflammation by secreting pro-inflammatory cytokines, while reactive anti-inflammatory microglia secrete anti-inflammatory mediators to reduce inflammation and promote tissue healing and repair. The role of macrophages and microglia in TMEV infection is intricate and not fully elucidated, with microglia exhibiting both beneficial and pathological roles during TMEV-IDD.

During demyelination, microglia and macrophages are found close to the lesion sites and contain TMEV viral antigens. Microglia and macrophages can efficiently take up viral particles by phagocytosis. While these cells have been shown to be persistently infected by TMEV in SJL mice, these cells are not highly permissive for viral replication, yielding few viral particles during TMEV infection [142,143,164,173,174]. Microglia constitutively express TLR1-9, and increased TLR expression can be induced during viral infection (reviewed in [55]). Microglia can sense a viral infection or cell-associated damage signals via PRRs and DAMPs, respectively, becoming reactive (Figure 2). Reactive inflammatory microglia trigger the expression of IFN-I, such as IFN-α and IFN-β, and NF-κB, inducing the production of inflammatory mediators including IL-6, IL-1b, IL-12, tumor necrosis factor (TNF)α, CCL2, CCL3, and CCL5 [42,64,175,176] (Figure 2).

IFN-I plays a critical role in the demyelinating disease following TMEV infection. The administration of IFN-β decreases demyelination in TMEV-IDD by reducing immune cell infiltration into the CNS, increasing the level of the anti-inflammatory cytokine IL-10 in the CNS, and reducing the myelin-specific CD4^+^ T cell response. In accordance with these data, blocking IFN-β increases demyelination [64]. In the BL6 mouse strain, which is resistant to TMEV-induced demyelination, a lack of IFN-β (IFN-β KO) resulted in impaired viral clearance, mild spinal cord demyelination, and sustained IFN-I-induced inflammation in BeAn-infected mice, while it was fatal for DA-infected mice [45]. This increased expression of inflammatory IFN-I-induced cytokines, such as Isg15, Tnfa, Il1a, and Il1b, could reflect the ongoing viral replication due to decreased antigen presentation in IFN-β-KO mice. Interestingly, the KO of IFN-β, specifically in neurons, oligodendrocytes, and astrocytes, did not increase viral replication, CNS inflammation, or hippocampal sclerosis, suggesting a role for microglia in infiltrating immune cells on an IFN-β-driven antiviral immune response and the resistance to TMEV persistence and demyelinating disease in BL6 mice [45]. Additionally, MDA5-deficient mice showed decreased INF-I production, particularly IFNα, increased viral loads, inadequate immune responses, and the development of a demyelinating disease in resistant 129/SvJ background mice, suggesting a protective role for MDA-5 in TMEV-IDD [177].

Reactive microglia upregulate MHC-I, MHC-II, and costimulatory molecules, becoming competent in presenting antigens to T cells [178,179,180]. Viral antigen presentation to Th1 CD4^+^ T cells leads to the release of chemokines by these T cells, increasing the recruitment of peripheral macrophages. The excess of CNS inflammation causes bystander myelin damage. A myelin antigen can be taken up by macrophages and microglia in TMEV-infected mice. Macrophages and microglia can present myelin to autoreactive myelin-specific CD4^+^ T cells [176]. Nevertheless, the exact role of the autoreactive CD4^+^ T cells in demyelination in TMEV-IDD remains not fully elucidated. The role of T cells in demyelination and disease progression is an active area of research, and current knowledge has been extensively reviewed elsewhere. Additionally, microglia, due to their phagocytic nature, are essential for clearing myelin debris to allow MS plaques to get remyelinated by oligodendrocytes [181,182,183].

To gain more insight into the exact role of microglia in demyelination during TMEV infection, microglia depletion experiments were performed. The chronic depletion of microglia using the small molecule inhibitor of colony-stimulating factor 1 (CSF1R), PLX5622, exacerbated TMEV-induced demyelination, similar to data obtained from microglia depletion in JHMV infection, another mouse model of viral-induced demyelination [184,185]. The chronic inhibition of CSF1R resulted in the accumulation of T cells in the CNS and increased the number of TMEV antigens in the spinal cord. Interestingly, repopulating microglia in PLX5622-depleted mice by discontinuing PLX5622 treatment at the onset of the disease development resulted in a decreased number of TMEV antigens in the spinal cord, as well as decreased mortality and weight loss. However, the clinical manifestation of the disease was not affected following microglia repopulation [186,187]. These data suggest an important protective role for microglia at the chronic phase of TMEV-IDD. It is important to note that PLX5622 treatment affects not only microglia, but also targets macrophages, as described by many research groups.

Interestingly, a recent study demonstrated the capacity of microglia to secrete exosomes containing viral RNA during TMEV infection [175]. The uptake of exosomes by uninfected bystander cells triggered the activation of innate immune signaling, leading to IFN-I, IL-6, IL-12, TNFα, and CCL2 secretion by these cells, and led to the initiation of the inflammatory response when transferred into naïve mice brains. This elegant study demonstrated a novel mechanism employed by microglia in inducing neuroinflammation [175]. However, it remains unknown if this is a mechanism of viral spread and persistence in TMEV-IDD.

Elevated numbers of infiltrating macrophages are found in MS lesions from patients and are associated with demyelination and axonal loss [188,189]. The reactive inflammatory levels of these cells exhibit variation depending on the stage of the disease [190]. Furthermore, myelin damage is correlated with the presence of reactive microglia and macrophages [191], and myelin-containing macrophages are also detected in active MS lesions [192]. These findings emphasize the critical role of infiltrating macrophages in disease development and the progression of the disease. In the context of inflammation, the innate antiviral factor IRF3 is found to be constitutively activated in macrophages from SJL mice, which can negatively impact IL-12 expression by increasing the levels of IL-12/23p40 and decreasing the levels of IL-12p35 during TMEV-IDD, affecting the Th1 immune response. Furthermore, like microglia, macrophages produce high levels of pro-inflammatory cytokines such as IL-6, IL-12, and inducible nitric oxide synthase (iNOS), with the last being associated with myelin and oligodendrocyte damage. The expression of IL-6 and IL-23 is important in inducing a Th17 immune response, which has been associated with chronic inflammation and autoimmune diseases.

Macrophage infiltration into the CNS relies heavily on the chemokine chemoattractant protein CCL2, also known as monocyte chemoattractant protein-1 (MCP-1) (Figure 2). The development of demyelination following TMEV infection correlates with an increase in the CCL2 expression [193,194,195]. CCL2 can be produced by many cells in the CNS, including neurons, astrocytes, and microglia, leading to the recruitment of peripheral macrophages into the CNS (Figure 2). Interestingly, treatment with anti-CCL2 in SJL mice infected with TMEV resulted in a decrease in macrophage infiltration into the CNS, concomitant with a reduced accumulation of effector cells, such as CD4^+^ T cells, and a decline in TMEV replication, ultimately leading to a significant decrease in the clinical manifestation of TMEV-IDD [105]. Similarly, in CCR2 KO SJL mice infected with TMEV, decreased demyelination and disease severity were observed with no differences in the frequency of CD4^+^ and CD8^+^ T cells in the CNS [196], suggesting that CCR2 signaling and macrophage recruitment into the CNS are important for disease development.

While microglia and blood-infiltrating macrophages are crucial contributors to demyelination induced by TMEV infection, distinguishing between these two populations and assessing their individual contributions to disease development has proven to be challenging. Notably, several disease-modifying drugs are utilized in the treatment of MS, with some impacting microglial and macrophage functions [197,198]. Interferon beta treatment, despite increasing the secretion of inflammatory cytokines by microglia, leads to the downregulation of MHC-II expression [197,199,200]. Glatiramer acetate alters the reactive state of these cells toward an anti-inflammatory profile [201]. Fingolimod treatment induces a decrease in the production of proinflammatory cytokines and is suggested to play a role in remyelination [197,202]. Interestingly, minocycline treatment, discussed later in this review, decreases the activation of microglia and macrophages, demonstrating a reduction in disease severity in preclinical models [203]. Additionally, clinical trials involving minocycline showed a beneficial impact on the modulation of MS [197,204,205]. While macrophages and microglia play critical roles in MS, there is no current treatment to exclusively target these cells. Given the plasticity of microglia and macrophages and their roles in both pathology and neuroprotection, further studies are needed to gain a deeper understanding of their involvement in demyelination, the interplay between these innate immune cells in the modulation of CD4^+^ and CD8^+^ T cell immune function during disease, and the development of therapeutic approaches to modulate and specifically skew these cells towards a neuroprotective function.

## 6. TMEV Infection in a Mouse Model of Viral-Induced Seizures and Epilepsy

More than 70 million people are living with epilepsy [206], a chronic, disabling neurological disorder that affects all age groups. Spontaneous recurrent seizures due to abnormal brain activity are a hallmark of epilepsy. It considerably affects the quality of life of patients [207] and is often associated with comorbidities and cognitive impairment. Epilepsy has many etiologies ranging from genetic, where more than 500 genes associated with epilepsy have been described, to acquired epilepsy, such as those developed after a traumatic brain injury (TBI) and a CNS infection [207,208]. Although there are available anti-seizure medications to control seizures, they only achieve 20–30% of seizure frequency reduction with a 30–50% responder rate. Furthermore, seizure freedom ranges from 2–5% only [209,210,211]. Despite treatment, 30% of patients continue to present with uncontrolled seizures [212]. While genetic mutations represent more than 50% of the underlying causes of epilepsy, CNS insults that trigger acute neuroinflammation, such as TBI, brain tumors, and CNS infections, significantly contribute to epilepsy development. The mechanisms of how neuroinvasion and neuroinflammation affect long-term changes in neuronal function remain poorly understood, therefore delaying progress in the development of effective treatments to prevent and cure epilepsy.

CNS viral infection is a significant contributor to the development of seizures and epilepsy [16,18,24,25,39,40,213]. Many viruses, including herpes simplex virus (HSV)-1, non-polio picornaviruses, Japanese encephalitis virus, West Nile virus (WNV), HHV-6, and SARS-CoV-2 (severe acute respiratory syndrome coronavirus 2) can cause encephalitis in humans. During acute viral encephalitis, patients have a 16-fold increased risk of experiencing seizures. Moreover, if seizures manifest during the acute phase of the infection, these patients have a staggering 22-fold higher chance of developing epilepsy compared to the general population [19,214]. To develop new treatment strategies for altering the course of disease development and preventing epilepsy in high-risk groups, it is essential to elucidate the mechanisms driving seizure development following viral encephalitis.

Animal models are a critical tool for studying epilepsy, and their use has extensively contributed to advancing our understanding of the pathophysiology of this disorder. While many researchers attempted to study seizures and epilepsy induced by viral infections in rodent models, these studies were limited due to the high mortality rate observed during the acute phase of the HSV-1 or WNV infection. However, when the murine virus TMEV was introduced via IC injection into the CNS of BL6/J mice, acute seizures and epilepsy were observed with a low mortality rate [215,216,217,218]. This is the first and only mouse model of viral-induced epilepsy, and its use has substantially increased our understanding of how viral infection and neuroinflammation contribute to the pathophysiology of seizures (Figure 1B) [33,46]. It is currently used in the Epilepsy Therapy Screening Program (ETSP) from the National Institute of Neurological Disorders and Stroke (NINDS) as a tool to identify compounds capable of reducing or blocking seizures [219]. This is a model of human TLE, the most common type of acquired epilepsy in adult human patients, and recapitulates many features observed in patients, such as hippocampal sclerosis, astrogliosis, microgliosis, the infiltration of macrophages from the periphery into the brain, and behavioral and cognitive alterations [220].

As mentioned earlier, SJL mice are considered permissive for chronic TMEV infection, while BL6 mice are resistant to TMEV-IDD but susceptible to seizures. In this model, an IC infection of BL6/J mice results in acute behavioral seizures that develop between 3 and 8 days pi [16,33,39,40,46,65,67] This is followed by a latent period where no seizures are observed, and TMEV is effectively cleared by the immune system around day 14. Between 30 and 90 days pi, spontaneous recurrent seizures are observed in 30–50% of the mice that experience acute seizures, and this is considered the epilepsy phase (Figure 1B). The incidence of acute seizures is dependent on the viral titer used for IC inoculation, which means the higher the viral titer/plaque-forming units (PFU) injected into the brain of these mice, the higher the incidence of seizures [65,221]. During the acute phase of the infection, TMEV has a tropism for the CA1 and CA2 regions of the hippocampus, and extensive hippocampal sclerosis is observed as early as 2 dpi [16,18,39,40,65,155,156,222]. CNS inflammation is associated with changes in the inhibitory and excitatory balance in neurons, promoting neuronal excitability and contributing to both ictogenesis and epileptogenesis [25,223,224,225,226,227,228,229,230].

### The Role of Microglia and Infiltrating Macrophages in Neuronal Excitation Following TMEV CNS Infection

Microglia and brain-infiltrating macrophages play distinct roles in seizure generation [65]. As previously mentioned, monocytes/macrophages typically reside in the periphery during homeostasis. However, in the context of a CNS viral infection, such as with TMEV, macrophages can infiltrate the brain parenchyma. After TMEV infection of neurons, these cells secrete inflammatory mediators and DAMPs due to cellular damage caused by the virus (Figure 2). The DAMPs and inflammatory mediators are sensed by resident microglia which respond to the stimuli and become reactive, releasing inflammatory cytokines and chemokines and migrating to the damaged site. Some of the inflammatory molecules secreted by microglia activate other glial cells, such as astrocytes, which can impact the permeability of the BBB. Moreover, certain molecules secreted by microglia, including CCL2, play a role in recruiting monocytes from the periphery into the brain (Figure 2). Notably, CCL2 has also been shown to be secreted by neurons following a TMEV infection [84]. In response to CCL2 and potential changes in the vasculature and BBB integrity, monocytes migrate from the periphery into the brain parenchyma where they differentiate into macrophages (Figure 2). Within the CNS, reactive inflammatory macrophages contribute to an escalation in CNS inflammation by secreting pro-inflammatory cytokines, like IL-6 (Figure 2). During the acute immune response to the TMEV infection, CD4^+^ and CD8^+^ T cells are also recruited into the CNS, playing a crucial role in TMEV clearance, typically occurring around day 14 pi.

Seizures are initially observed around day 3 pi and are accompanied by the infiltration of peripheral immune cells, including neutrophils, macrophages, NK cells, B cells, and T cells [66,231]. Macrophage infiltration into the brain of TMEV-infected mice can be detected within hours post-infection [155]. Interestingly, despite a significant number of T cells infiltrating the brain during TMEV infection, TMEV infection of mice lacking T or B cells (RAG^−/−^) showed a similar seizure incidence compared to the wild-type (WT) group [67]. A lack of a correlation between T cell infiltration and seizure development was also previously suggested by another group [232], supporting the hypothesis that seizures occur due to the activation of the innate immune response, which precedes the induction of the adaptive immune response. To test this hypothesis, TMEV-infected mice were treated with NK, neutrophil-blocking antibodies, or a CXCL2 inhibitor, which prevent neutrophils from migrating into the CNS. Interestingly, none of these treatments altered the incidence of seizures in TMEV-infected mice, suggesting that lymphocytes, neutrophils, and NK cells do not play a critical role in the generation of acute seizures [231]. Additionally, mice treated with minocycline, an antibiotic with anti-inflammatory properties, showed reduced microglia activation, reduced immune cell infiltration from the periphery into the CNS, and a decreased seizure incidence [66]. However, in a recent study, minocycline treatment resulted in no effect on the seizure incidence [219]. This discrepancy in the results could be due to differences in drug treatment protocols. In both studies, minocycline was administered twice a day at a concentration of 50 mg/kg. However, a reduction in the seizure incidence was specifically observed when minocycline treatment started 24 h before TMEV infection and was continued daily for 7 days [66]. A loss of seizure control occurred when minocycline treatment was given from 3 to 7 dpi [219]. These findings suggest that the early control of inflammation is crucial for modulating seizures in TMEV-infected mice. Interestingly, another study demonstrated that minocycline treatment improved long-term behavioral comorbidities [233]. Importantly, when BL6/J mice infected with TMEV were pharmacologically depleted of macrophages using clodronate liposomes, the incidence of seizure was drastically reduced, highlighting the significant role of macrophages in the pathogenesis of acute seizures following TMEV infection [63,67].

The infiltration of macrophages is an early and prominent feature during an acute TMEV infection, and these cells are the major producers of IL-6 in the brains of TMEV-infected mice. IL-6 KO mice infected with TMEV show a significant decrease in seizure incidence compared to WT mice. Interestingly, TMEV H101, a TMEV-DA mutant that does not replicate within the brain parenchyma when administered via IC inoculation, causes seizures in 40% of infected mice. Intriguingly, higher levels of IL-6 were found in the periphery of TMEV H101-infected mice, suggesting that pathological IL-6 levels outside the CNS may contribute to seizure generation [234]. IL-6 is a pleiotropic cytokine that is abundantly present in the serum, cerebrospinal fluid, and tissues of patients with epilepsy and refractory epilepsy [235,236]. While the convulsive impact of IL-6 has been recognized, the precise mechanisms of how IL-6 contributes to neuronal excitation remain incompletely understood. It has been proposed that the IL-6 modulation of GABAergic inhibition may play a role in the IL-6-induced neuronal excitation [39,234,237,238,239,240]. Clinical studies on the immune-modulatory effect of Tocilizumab, an IL-6 receptor (IL-6R) antagonist that prevents the binding of IL-6 to the IL-6R, yielded promising results as a potential adjuvant for the treatment of acute seizures and refractory epilepsy [241,242,243,244].

IL-6 can signal through two different pathways: classic and trans-signaling. Classic signaling involves the binding of IL-6 to the cell membrane-associated IL-6 receptor (mIL-6R), triggering the homodimerization of the glycoprotein 130 (gp130). In the trans-signaling pathway, soluble IL-6R (sIL-6R), generated by IL-6R cleavage from the membrane (by the proteases ADAM [Disintegrin and Metalloproteinase] 10 and/or ADAM 17) or via alternative splicing (in humans), binds to the secreted IL-6. The sIL-6R-IL-6 complex binds to and activates gp130 [245,246]. While IL-6R expression is cell-type-restricted, gp130 is expressed ubiquitously; therefore, classic IL-6 signaling is restricted to IL-6R-expressing cells, whereas trans-signaling can occur in any cells. Both pathways lead to the phosphorylation of STAT3, but trans-signaling exhibits a more sustained and higher amplitude response. Notably, classic signaling is associated with protection and anti-inflammatory properties, while trans-signaling exhibits pathological and pro-inflammatory roles [245,246]. Thus, it is important to recognize that IL-6R inhibitors cannot differentiate between classic and trans-signaling, blocking both pathways and potentially leading to unexpected and detrimental side effects.

The infiltration of macrophages relies on CCL2-CCR2 signaling, as evidenced by impaired monocyte/macrophage migration in CCR2 KO mice responding to CCL2 secretion [76]. CCL2 can be produced by neurons, microglia, astrocytes, and endothelial cells within the CNS, and CCL2 expression is highly induced in the brains of patients with epilepsy as well as in the brains of animal models of epilepsy [16,247,248,249]. Intriguingly, TMEV-infected CCR2 KO mice showed a significant reduction in macrophage infiltration into the CNS [18,250]. However, the seizure incidence and severity remained similar in CCR2 KO and WT mice, suggesting a possible role for the elevated cytokine levels within the CNS as potential seizure triggers. Additionally, CCR2 KO mice also demonstrated a significant decrease in hippocampal sclerosis [18,250], suggesting that the neuronal death of hippocampal neurons occurs independently of the TMEV infection of the neuron, as previously suggested, pointing to a central role of immune cells in hippocampal neuronal damage [18,155,158]. Interestingly, CCR2 KO mice infected with TMEV have fewer viral antigens at 7 days pi compared to WT mice [18,250], which may explain the decreased hippocampal sclerosis in these mice. Importantly, while clodronate-treated mice showed significant seizure reduction, the development of hippocampal sclerosis was not prevented [63,67]. Similarly, while macrophage depletion using clodronate liposomes reduces seizure incidence [63,67], inhibiting macrophage infiltration into the brain in CCR2 KO mice shows no effect on seizures. This discrepancy between pharmacological and genetic approaches highlights the need for further studies to unravel this puzzle.

Reactive microglia during a CNS viral infection triggers the production of pro-inflammatory cytokines, such as IL-1β and TNF-α, leading to increased neuronal excitation and a decreased seizure threshold. While TMEV infection of mice deficient in IL-1R1 showed no significant difference in the seizure incidence compared to control mice, TNFR1 KO mice showed a notable reduction in the occurrence of seizures [231]. TNFR1 can be activated by both soluble (sTNF-α) or membrane-bound (mTNF-α) TNF-α [251,252]. The treatment of TMEV-infected mice with XPro1595, a dominant negative inhibitor of sTNF-α, resulted in no significant effect on the seizure incidence and severity [68]. One plausible explanation for the lack of efficacy in seizure control following XPro1595 treatment is that while TNFR1 is preferentially activated by sTNF-α, other ligands, such as tmTNFa, which are not affected by XPro1595 treatment, can still activate TNFR1 [68,251,252], thereby contributing to neuroinflammation and seizure generation.

Initial studies implicated microglia as the primary TNF-α producer during the acute seizure phase, yet depleting microglia had no impact on the levels of TNF-α, suggesting the involvement of other cell types in TNF-α secretion, with macrophages emerging as significant contributors [187]. TNF-α is highly induced in the hippocampus of TMEV-infected mice during acute seizures and modulates the trafficking of hippocampal glutamate receptors, specifically via TNFR1, contributing to neuronal hyperexcitation [68].

In an effort to determine the role of microglia in seizures, mice were treated with the CSF1R inhibitor PLX5622. Surprisingly, the seizure incidence remained the same as compared to the controls, although increased seizure severity was reported in the PLX5622-treated group [187,253]. Notably, microglia-depleted mice experienced fatal viral encephalitis, increased neuronal loss, earlier seizures, and spinal cord neurodegeneration. Plaque assay and RT-qPCR found increased levels of TMEV in the CNS of TMEV-infected mice treated with PLX5622, suggesting a protective role for microglia in orchestrating the antiviral immune response. Intriguingly, although macrophages migrated into the brain of TMEV-infected mice treated with PLX5622, they were unable to compensate for the loss of microglia in terms of effective viral clearance [187,253], suggesting microglia have a specialized role in controlling viral infections of the CNS. In a mouse model of kainic acid-induced status epilepticus, microglia depletion aggravated the severity of seizures [254], emphasizing the beneficial role of microglia in controlling the development of seizures. As highlighted earlier, CSF1R is expressed in macrophages, and its inhibition can decrease macrophage numbers and alter their polarization and function [255,256], which may influence the antiviral response during a neurotropic viral infection.

## 7. Conclusions and Future Perspectives

In this review, we explored the current understanding of the intricate relationship between microglia and CNS-infiltrating macrophages in the context of mouse models for viral-induced MS and epilepsy pathogenesis. Despite advancements in the field of neuroinflammation and neurodegeneration research, the precise mechanisms underlying the impact of these two cell types on neuronal excitation and demyelination remain elusive. Both microglia and macrophages emerge as pivotal players in disease development. Interestingly, microglia exhibit a dual role, serving both protective and pathological functions in TMEV-induced seizures and demyelination. Conversely, brain-infiltrating macrophages appear to predominantly contribute to disease pathology. Depleting macrophages correlates with reduced demyelination and diminished seizures in TMEV-infected SJL or BL6/J mice, respectively, suggesting a potential avenue for novel therapeutic interventions. Nevertheless, caution is necessary when targeting microglia therapeutically, given their crucial antiviral, neuroprotective, and neuromodulatory roles.

Still, it is crucial to advance our understanding through the development of methods to differentiate between microglia and brain-infiltrating macrophages during neuroinflammatory conditions. Despite the substantial amount of bulk and scRNA transcriptomics data on microglia and macrophages in CNS pathologies and the acknowledgment of their heterogeneity, there remains a lack of space resolution regarding the phenotypic state during disease development and progression. This analytical gap prevents us from obtaining unbiased spatiotemporal information on dynamic changes in gene expression, limiting our comprehension of the functional roles played by microglia and macrophages during neurotropic viral infections. Addressing this gap will significantly advance our knowledge and pave the way for the development of more targeted therapeutical interventions.

## Figures and Tables

**Figure 1 viruses-16-00119-f001:**
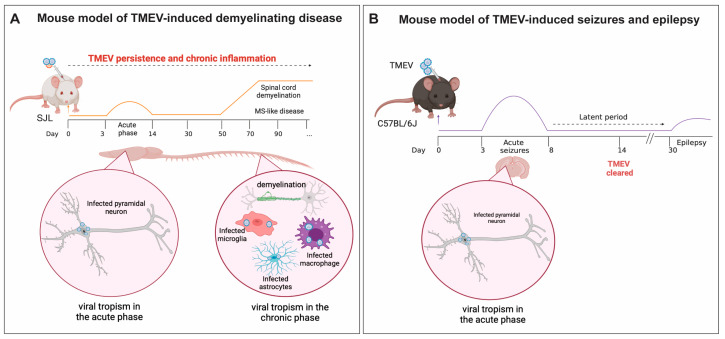
**Schematic representation of** (**A**) Mouse model of Theiler’s murine encephalomyelitis virus-induced demyelination disease (TMEV-IDD): Intracerebral (IC) infection of SJL/J mice with TMEV results in a persistent viral infection throughout the mouse’s lifespan. This is a biphasic disease; an acute phase occurs between 3 and 14 dpi; chronic neuroinflammation due to viral persistence in the central nervous system (CNS) leads to a multiple sclerosis (MS)-like phase, which appears 40–60 days post-infection. In this phase, weakness of the hind limbs and ataxic paralysis are observed. During the acute phase, TMEV primarily infects CA1 and CA2 hippocampal neurons and leads to the activation of innate and adaptive immune responses. Due to failure in viral clearance, chronic neuroinflammation is observed during MS-like disease and is characterized by viral persistence in the white matter of the spinal cord, and oligodendrocytes (not shown here), astrocytes, microglia, and infiltrating macrophages are the main cells infected by TMEV during this phase. (**B**) Mouse model of viral-induced seizures and epilepsy. C57BL/6J mice ICinfected with TMEV develop encephalitis. Between 3 and 8 dpi, mice develop acute seizures, followed by a latent period in which seizures are no longer observed. Between 30 and 100 dpi, a portion of the mice that experienced acute seizures develop spontaneous recurrent seizures (epilepsy). During the acute phase of the infection, TMEV infects pyramidal neurons in the CA1 and CA2 regions of the hippocampus. Induction of innate and adaptive immune response results in viral clearance, but also contributes to neuronal excitation and seizure development. Figure made using Biorender.

**Figure 2 viruses-16-00119-f002:**
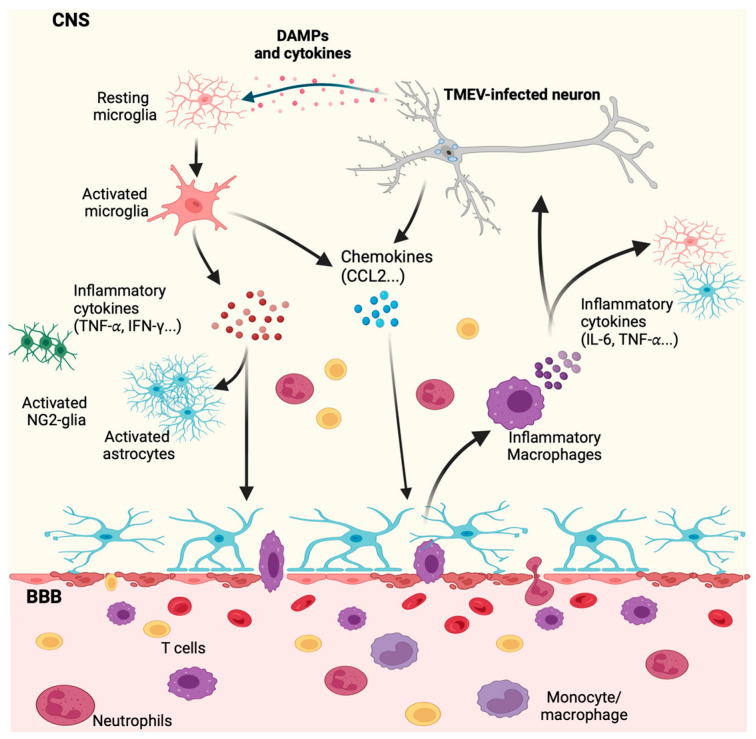
**Innate immune response in the CNS during neurotropic TMEV infection**. Infection of neurons by TMEV results in neuronal damage and secretion of cytokines by these cells. Damage-associated molecular patterns (DAMPs) and cytokines are sensed by microglia, leading to microglial adopting a reactive inflammatory state, causing microglia to secrete pro-inflammatory cytokines. Pro-inflammatory cytokines such as tumor necrosis factor (TNFα) can activate and amplify the reactive state of astrocytes, contributing to the loss of the blood-brain barrier (BBB) integrity. Additionally, infected neurons and reactive inflammatory microglia secrete chemokines, leading to the recruitment of leukocytes (T cells, neutrophils, and monocytes/macrophages) from the periphery into the CNS. C-C motif ligand 2 (CCL2) secretion plays a role in monocyte/macrophage infiltration into the brain. Once in the brain, these inflammatory macrophages produce pro-inflammatory cytokines, that can act on neuronal and glial cells, exacerbating neuroinflammation and contributing to the CNS pathology. Figure made using Biorender.

**Figure 3 viruses-16-00119-f003:**
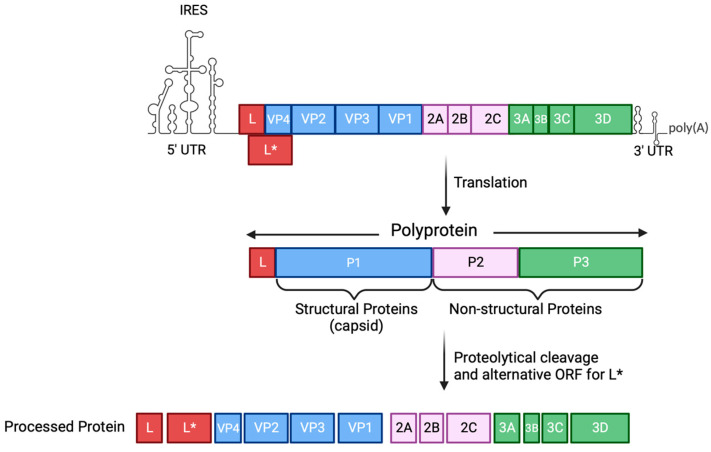
**Schematic representation of the TMEV genome organization.** The positive-sense RNA molecule from TMEV is flanked by 5′ and 3′ UTRs. TMEV genome is translated into a polyprotein which, through proteolytic cleavage by the 3C protease, gives rise to 12 proteins (4 structural and 8 non-structural proteins). An additional protein (L*) is expressed from an alternative ORF. IRES, internal ribosome entry site; UTR, untranslated region; ORF, open reading frame. Figure made using Biorender.

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
