# Peer review of "The Contribution of Microglia and Brain-Infiltrating Macrophages to the Pathogenesis of Neuroinflammatory and Neurodegenerative Diseases during TMEV Infection of the Central Nervous System"

_viruses, 2024, doi:10.3390/v16010119_

Round 1
Reviewer 1 Report
Comments and Suggestions for Authors
This is an excellent review and there are no serious criticisms regarding the information presented and its interpretation.
Minor points:
- I think "referred to as TMEV henceforth3. TMEV infection" on line 102 is a typo.
- The titles of papers listed in the References are a mixture of some with capital letters at the beginning of each word and others with lowercase letters.
Author Response
Dear reviewer,
Thank you for taking the time to review this manuscript and for providing such positive feedback. Your minor changes were addressed and can be found highlighted in the revised submitted file.
Reviewer 2 Report
Comments and Suggestions for Authors
Suggestions for edits:
1. Introduction – line 43-44: please add citations for evidence of viral infections playing a role in stated diseases – MS, AD, epilepsy, etc. This would help your reader and increase the value of this review.
2. It would be useful to a non-expert to further discuss why spinal cord demyelination is observed in one model versus hippocampal sclerosis in another? It is known that SJL vs B6 mice exhibit different disease phenotypes but why there are regional differences with CNS infection is unclear. Are SJL mice infected at the spinal column?
3. Line 181 – suggest changing to “Differentiating “in the setting of CNS inflammation”’ rather than “during CNS inflammation”.
4. Section 4.1 – There is a lot of information on the immune signaling pathways that would be well served with a figure for reference by a non-expert. Please refer more frequently to your figures, in general. Additionally, if the figures can be placed in context with the relevant section, it would be beneficial to guide a reader.
5. Section 5 – It is stated that MS incidence is “on the rise”. Is there a particular population or region where this is most notably occurring and if so, please clarify.
6. Lines 326-328 are redundant with lines 380-382. Please consider conciseness in the text themes as there is a lot of information presented in this article.
7. Lines 492-500: Any clinical evidence for a role of microglia or macrophages in human MS patients? Any current treatments that act on these targets? Please further contextualize the translational/therapeutic potential for this section.
8. Lines 592-600: Please further discuss the relevant impact of dose and frequency of MIN administration. I believe that these two discussed studies employed different dosing regimens, which may have impacted the outcomes. Speculation on how subtle modulation of microglial activation state would be useful.
9. Lines 661-663: Please discuss why TNFR1 inhibition with XPro did not demonstrate any beneficial effects on acute seizures in TMEV model (PMID 28497109)?
10. Figure 2 is not sufficiently referred to in the text body. Please consider making it more integral to the manuscript text to warrant placement.
Author Response
Dear reviewer,
Thank you for taking the time to review this manuscript. I have tried to address reviewers’ concerns to the best of my ability. The changes and answers to your questions are addressed below and highlighted in the revised submitted file.
- Introduction – line 43-44: please add citations for evidence of viral infections playing a role in stated diseases – MS, AD, epilepsy, etc. This would help your reader and increase the value of this review.
Citations were added.
- It would be useful to a non-expert to further discuss why spinal cord demyelination is observed in one model versus hippocampal sclerosis in another column?
Spinal cord demyelination occurs as a result of chronic inflammation due to viral persistence, and this is often mentioned throughout the manuscript. Hippocampal sclerosis is thought to occur due to a fast induction of the innate response. Thank you for your suggestion of adding this to the review. The following paragraph was added to explain the differences in hippocampal sclerosis between SJL and BL6 (line 380): “In contrast to BL6 mice, in SJL-infected mice most of the hippocampal neurons are preserved, showing mild hippocampal sclerosis (Howe, Lafrance-Corey et al. 2012, Howe, Lafrance-Corey et al. 2012). This variation in hippocampal sclerosis is not attributed to differences in viral titer or viral tropism between BL6 and SJL mice. Instead, it is potentially a consequence of the varying intensity of the innate immune response elicited by these two distinct mouse strains(Buenz, Rodriguez et al. 2006, Buenz, Sauer et al. 2009, Howe, Lafrance-Corey et al. 2012). Interestingly, during the acute phase of TMEV infection in SJL, a transient increase in IL-10-related genes has been observed (Uhde, Ciurkiewicz et al. 2018). IL-10 is an anti-inflammatory cytokine that plays a critical role in controlling the immune response and inflammation. Inhibition of the IL-10 signaling in TMEV-infected SJL resulted in enhanced neuronal loss and hippocampal damage, emphasizing once again the significance of amplitude of the immune response in neuronal damage (Uhde, Ciurkiewicz et al. 2018)”
- Line 181 – suggest changing to “Differentiating “in the setting of CNS inflammation”’ rather than “during CNS inflammation”.
Answer: The change was made and now reads (line 184): “Differentiating between microglia and macrophages in the setting of CNS inflammation, although crucial, has been challenging.”
- Section 4.1 – There is a lot of information on the immune signaling pathways that would be well served with a figure for reference by a non-expert. Please refer more frequently to your figures, in general. Additionally, if the figures can be placed in context with the relevant section, it would be beneficial to guide a reader.
Answer: Figures were referenced more often and placed in the context of relevant sections. Figures will be placed in the context of relevant sections.
- Section 5 – It is stated that MS incidence is “on the rise”. Is there a particular population or region where this is most notably occurring and if so, please clarify.
The following was added to address the comment:
Lines 303 – 309: “Worldwide, 2.8 million people live with MS [122]. While MS exhibits distinct geographic prevalence, with higher incidence in Europe and North America and lower rates in sub-Saharan Africa and East Asia, its overall prevalence is increasing globally [122]. In only 14% of the reporting countries, MS incidence has remained stable or decreased. There has also been substantial growth in the pediatric onset of MS was also reported [122, 123]. While the reason for this increase is currently unknown, genetic and environmental factors may be associated with this trend [124].”
- Lines 326-328 are redundant with lines 380-382. Please consider conciseness in the text themes as there is a lot of information presented in this article.
Answer: This concern was resolved. Redundancy was removed from lines 326-328. The following sentence was added (lines 335 – 336): “Many mechanisms have been proposed to explain the correlation between viral infection and the development of autoimmune diseases.”
- Lines 492-500: Any clinical evidence for a role of microglia or macrophages in human MS patients? Any current treatments that act on these targets? Please further contextualize the translational/therapeutic potential for this section.
Reactive microglia and macrophages, along with the presence of phagocytosed myelin within these cells, are commonly found in lesions from MS patients and in demyelinating lesions of preclinical models. The modulation of these cells in preclinical model settings modulates disease development, demonstrating their active involvement in the pathogenesis of MS.
The following has been revised and included to address your comment:
Lines 488-493: “The reactive inflammatory levels of these cells exhibit variation depending on the stage of the disease [190]. Furthermore, myelin damage is correlated with the presence of reactive microglia and macrophages {Luo, 2017, 28721047} and myelin-containing macrophages are also detected in active MS lesions {Grajchen, 2018, 30454040}. These findings emphasize the critical role of infiltrating macrophages in disease development and progression of the disease”.
Lines 518 – 529: Notably, several disease-modifying drugs are utilized in the treatment of MS, with some impacting microglial and macrophage functions [197, 198]. Interferon beta treatment, despite increasing the secretion of inflammatory cytokines by microglia, leads to downregulation of MHC-II expression [197, 199, 200]. Glatiramer acetate alters the reactive state of these cells toward an anti-inflammatory profile [201]. Fingolimod treatment induces a decrease in the production of pro0inflammatory cytokines and is suggested to play a role in remyelination [197, 202]. Interestingly, minocycline treatment, discussed later in this review, decreases the activation of microglia and macrophages, demonstrating a reduction in disease severity in preclinical models [203]. Additionally, clinical trials involving minocycline showed a beneficial impact on the modulation of MS [197, 204, 205]. While macrophages and microglia play critical roles in MS, there is no current treatment to exclusively target these cells.
- Lines 592-600: Please further discuss the relevant impact of dose and frequency of MIN administration. I believe that these two discussed studies employed different dosing regimens, which may have impacted the outcomes. Speculation on how subtle modulation of microglial activation state would be useful.
Answer: This is an important and relevant point. Thank you for this question. The discussion was included, and it reads as follows (lines 618 – 624): “ This discrepancy in the results could be due to differences in drug treatment protocols. In both studies, minocycline was administered twice a day at a concentration of 50mg/Kg. However, a reduction in seizure incidence was specifically observed when minocycline treatment started 24 hours before TMEV infection and was continued daily for 7 days {Cusick, 2013, 23236075}. A loss of seizure control occurred when minocycline treatment was given from 3 to 7 dpi {Metcalf, 2022, 34668659}. These findings suggest that early control of inflammation is crucial for modulating seizures in TMEV-infected mice”
- Lines 661-663: Please discuss why TNFR1 inhibition with XPro did not demonstrate any beneficial effects on acute seizures in TMEV model (PMID 28497109)?
This is also an interesting point. Thank you for this comment and following discussion was added (lines 700 – 707): “TNFR1 can be activated by both soluble (sTNF-?) or membrane-bound (mTNF-?) TNF-? [251, 252]. Treatment of TMEV-infected mice with XPro1595, a dominant negative inhibitor of sTNF-?, resulted in no significant effect on seizure incidence and severity [68]. One plausible explanation for the lack of efficacy in seizure control following XPro1595 treatment is that while TNFR1 is preferentially activated by sTNF-?, other ligands, such as tmTNFa, which is not affected by XPro1595 treatment, can still activate TNFR1 [68, 251, 252], thereby contributing to neuroinflammation and seizure generation.”
Figure 2 is not sufficiently referred to in the text body. Please consider making it more integral to the manuscript text to warrant placement.
Answer: Figure 2 was referred more in the relevant sessions.
Round 2
Reviewer 2 Report
Comments and Suggestions for Authors
All my prior comments have been adequately addressed.